# The proportion of Thai postmenopausal women who would be eligible for anti-osteoporosis therapy

Piyachat Chanidkul[1], Dueanchonnee Sribenjalak[1], Nipith Charoenngam[2,3], Chatlert Pongchaiyakul[1] *

1 Division of Endocrinology and Metabolism, Department of Medicine, Faculty of Medicine, Khon Kaen University, Khon Kaen, Thailand, 2 Department of Medicine, Mount Auburn Hospital, Harvard Medical School, Cambridge, Massachusetts, United States of America, 3 Department of Medicine, Faculty of Medicine Siriraj Hospital, Mahidol University, Bangkok, Thailand

* pchatl@kku.ac.th

**Data Availability Statement:** All relevant data are within the paper and its Supporting Information files.

## Abstract

### Purpose

To determine the proportion of postmenopausal Thai women who would be classified as having high risk of fracture and eligible for anti-osteoporosis therapy according to the National Osteoporosis Foundation (NOF) criteria.

### Methods

Postmenopausal Thai women aged 40–90 years who had been screened for osteoporosis during 2014–2019 were recruited. Demographic data and osteoporosis risk factors were collected based on the Fracture Risk Assessment Tool (FRAX) questionnaire. Bone mineral density (BMD) at the femoral neck and lumbar spine measured using dual energy X-ray absorptiometry. Ten-year probabilities of hip and major osteoporotic fracture (MOF) were calculated based on the Thai FRAX model with BMD. The study's protocol was approved by the Institutional Ethical Committee (HE581241).

### Results

A total of 3,280 postmenopausal women were included. The mean ± SD age was 63.6 ± 10.1 years. A total of 170 (5.2%) participants had a history of hip and/or vertebral fracture. After excluding these participants with fracture history, 699 (21.3%) had osteoporosis, 355 (10.8%) had osteopenia with high risk of fracture (FRAX 10-year probability of hip fracture $\geq$ 3% and/or MOF $\geq$ 20%), 1192 (36.3%) had osteopenia with low risk of fracture (FRAX 10-year probability of hip fracture < 3% and MOF < 20%) and 864 (26.3%) had normal BMD. Taken together, a total of 1,224 (37.3%) participants would be eligible for anti-osteoporosis therapy (prior fracture, osteoporosis or osteopenia with high risk of fracture).

**Funding:** The authors received no specific funding for this work.

**Competing interests:** The authors have declared that no competing interests exist.

## Conclusion

The prevalence of Thai postmenopausal women who would be eligible for anti-osteoporosis therapy was 37.3%.

## Introduction

Fragility fractures are fractures that result from low energy forces and are major clinical manifestations of osteoporosis. Fracture carries a significant public health burden because it is associated with patients' quality of life, substantial pain and disability, increased risk of morbidity and mortality and health-care costs [1–4]. To illustrate the burden of this condition, it has been shown that individuals with a history of fracture have a two-fold increased risk of mortality [5]. In addition, presence of a fragility fracture results in an increased risk of development of subsequent fractures, particularly in the first two years after the initial fracture, also known as an imminent fracture risk [6, 7]. The burden of fragility fractures is expected to be more pronounced in many regions of the world given the predicted increased number of aging populations in many countries [8, 9].

Given the significant burden of fracture, attempts have been made to determine the risk of fracture based on clinical information and bone mineral density (BMD) with the aim to guide the decision to pursue interventions to prevent subsequent fragility fractures. The fracture risk assessment tool, also known as FRAX, is a model that incorporates 12 risk factors developed to determine 10-year risk of hip fracture and major osteoporotic fracture (MOF), which has been calibrated in several countries [10–13]. The US National Osteoporosis Foundation (NOF) [3] has defined recommended that individuals with high-risk fracture receive anti-osteoporosis therapy. These individuals include postmenopausal women and men aged 50 years and older who have a prior fracture, osteoporosis (T-score of ≤ -2.5 at the femoral neck and/or lumbar spine), or osteopenia (T-score -1 to -2.5) with a FRAX 10-year probability of at least 3% for hip fracture or at least 20% for major osteoporotic fracture [5].

The prevalence of individuals who met the NOF criteria for anti-osteoporosis therapy has been reported in a few studies in Western population [14, 15]; however, these data in the Southeast Asian populations are relatively limited [16]. Therefore, we aimed to determine the proportion of postmenopausal Thai women who would be classified as having a high risk of fracture and eligible for a pharmacological treatment according to the NOF criteria.

## Methods

The current study was designed as a descriptive study in Srinagarind Hospital, a tertiary setting in Northeast of Thailand between 2010 and 2019. Postmenopausal Thai women aged 40 to 90 years who had been screened for osteoporosis between Jan 2014 and February 2019 were recruited. Patients with medication-induced or surgical menopause were excluded from the study. After completing the informed consent, participants were interviewed by a well-trained nurse to ascertain their demographic data and osteoporosis risk factors for FRAX questionnaire. Prior fractures including hip and vertebral fractures using the ICD-10 diagnoses and self-report were reviewed and recorded. In this study only fragility fractures were included. A review of the ICD-10 diagnoses from the hospital database, as well as self-report and the medical record and film X-ray including hip and vertebral fracture. Body weight (including light indoor clothing) was measured using an electronic scale (accuracy of 0.1 kilogram) and

standing height (without shoes) was measured using a stadiometer. Body mass index (BMI) was derived as the weight in kilograms divided by the square of the height in meters ($kg/m^2$).

BMD at the FN and LS were measured using dual energy X-ray absorptiometry on a Lunar Prodigy bone densitometer (GE Healthcare, Madison, WI, USA). BMD T-scores were analyzed using Asian population reference databases, supplied by the manufacturer. The coefficient of variation for BMD for normal participants ranged from 1.3–1.5% and 1.5–2.0% for FN and LS, respectively. FRAX scores with femoral BMD were calculated using an online calculator for each individual in the study (https://www.sheffield.ac.uk/FRAX/tool.aspx?country=9) based on the Thai reference. Clinical data and femoral neck T-score data were input into the Thai FRAX model to obtain the 10-year risk of hip and major osteoporotic fractures [11]. The study's protocol was approved by the Human Research Ethics Committee of Khon Kaen University (HE581241).

## Statistical analysis

Data were mainly analyzed by descriptive statistical methods. Mean and standard deviation (SD), and proportions for continuous and categorical variables were presented, respectively. Based on the femoral neck and lumbar spine T-score data, the prevalence of osteoporosis (T-score $\leq$ -2.5) was calculated. Based on NOF's recommendation for osteoporosis treatment, we determine the proportion of individuals who had prior fracture (fragility fracture, i.e., hip, vertebral fracture), osteoporosis at femoral neck and/or lumbar spine, and osteopenic individuals who had a 10-year probability of hip $\geq$ 3% and/or MOF $\geq$ 20%. In addition, we determined the proportion of individuals at high risk by using a 10-year probability of MOF $\geq$ 10% in this study [17]. Comparison of dependent variables between study groups were made using independent t-test or ANOVA as appropriate. The Chi-square analysis was used to compare categorical variables between groups. The odds ratio (OR) with 95% confidence interval (CI) were used to determine the association between osteoporosis at FN/LS and clinical characteristics (fractures and 10-yr probability of hip and MOF). The correlation among variables were analyzed using the Pearson's correlation. Statistical significance was defined as p-value of <0.05. All statistical analyses were performed using SPSS version 19 (SPSS Inc, Chicago, IL, USA.).

## Results

A total of 3,280 participants were recruited in the study. The mean ± SD age and BMI were ~63.6 ± 10.1 years and 23.9 ± 3.9 $kg/m^2$, respectively. A total of 170, 1156, 1103 and 851 patients aged 40 –<50, 50 –<60, 60–70 and >70 years, respectively. The prevalence of osteoporosis is 23.0%, with 12.4% and 18.6% of the participants having osteoporosis (T-score BMD <-2.5) at FN and LS, respectively. **Table 1** demonstrates characteristics of participants with and without osteoporosis. Compared with participants without osteoporosis, postmenopausal women with osteoporosis were older and had lower BMI, FN and LS BMD, and higher 10-year probability of hip fracture and major osteoporotic fracture with and without BMD (**Table 1**).

**Table 2** demonstrates FB and LS BMD and FRAX scores by age group (<60 years, 60–70 years vs >70 years). The prevalence of osteoporosis and hip fracture as well as 10-year probability of hip fracture and MOF with and without BMD increased significantly with age (p<0.001). However, the proportion of participants with a history of all fracture or vertebral fracture did not differ among age groups (**Table 2**). Correlation analysis revealed that increased age was positively associated with increased 10-year probability of hip fracture

**Table 1. Characteristics of participants with and without osteoporosis.**

| | All participants | Osteoporosis at FN and/or LS | Non-Osteoporosis | p-value |
|---|---|---|---|---|
| | (N = 3280) | (N = 753) | (N = 2527) | |
| Age (years) | 63.6 ± 10.1 | 70.6 ± 9.7 | 61.5 ± 9.2 | <0.001 |
| Body Weight (kg) | 56.0 ±9.9 | 49.6 ± 8.5 | 57.9 ± 9.5 | <0.001 |
| Height (cm) | 152.9 ± 6.1 | 150.1 ± 6.4 | 153.7 ± 5.7 | <0.001 |
| Body mass index (kg/m$^2$) | 23.9 ± 3.9 | 22.0 ± 3.5 | 24.5 ± 3.8 | <0.001 |
| FN BMD (g/cm$^2$) | 0.769 ± 0.137 | 0.624 ± 0.094 | 0.812 ± 0.117 | <0.001 |
| LS BMD (g/cm$^2$) | 0.966 ± 0.182 | 0.755 ± 0.114 | 1.028 ± 0.148 | <0.001 |
| 10-year probability of hip fracture with BMD | 2.52 ± 3.59 | 6.04 ± 5.34 | 1.48 ± 1.85 | <0.001 |
| 10-year probability of major osteoporotic fracture with BMD | 7.09 ± 5.47 | 12.27 ± 7.08 | 5.54 ± 3.68 | <0.001 |
| 10-year probability of hip fracture without BMD | 2.60 ± 3.36 | 4.95 ± 0.16 | 1.90 ± 0.05 | <0.001 |
| 10-year probability of major osteoporotic fracture without BMD | 6.94 ± 5.04 | 10.24 ± 5.82 | 5.96 ± 4.33 | <0.001 |

Abbreviations: BMD: Bone mineral density; FN: Femoral neck; LS: Lumbar spine

(r = 0.50 and 0.573, p<0.001) and MOF (r = 0.573, p<0.001) and was negatively associated with FB and LSBMD (r = -0.522 and -0.318, p<0.001 at FN and LS, respectively).

Table 3 demonstrates proportion of participants with prior history of fracture and FRAX score status among participants with and without osteoporosis. Compared with participants without osteoporosis, postmenopausal women with osteoporosis had higher likelihood of history of fracture (OR 1.61, 95%CI 1.15–2.24, p<0.05), hip fracture (OR 5.55, 95%CI 3.02–10.20, p<0.001) as well as FRAX scores of hip fracture ($\geq$ 3%) (OR 14.27, 95%CI 11.77–17.30) and/ or MOF ($\geq$ 10% or 20%) (OR 13.98, 95%CI 11.53–16.95 for MOF $\geq$ 10% and OR 14.27, 95%

**Table 2. Age-stratified bone mineral density and fracture risk assessment score status.**

| | Age group | | | | p-value |
|---|---|---|---|---|---|
| | <60 (N = 1326) | 60–70 (N = 1103) | >70 (N = 851) | All ages (N = 3280) | |
| FN BMD (g/cm$^2$) | 0.839 ± 0.13 | 0.757 ± 0.11 | 0.674 ± 0.12 | 0.769 ± 0.14 | <0.001 |
| LS BMD (g/cm$^2$) | 1.023 ± 0.16 | 0.955 ± 0.16 | 0.890 ± 0.19 | 0.965 ± 0.18 | <0.001 |
| 10-year probability of hip fracture with BMD | 0.82 ± 1.71 | 2.47 ± 3.49 | 5.25 ± 4.17 | 2.52 ± 3.59 | <0.001 |
| 10-year probability of major osteoporotic fracture with BMD | 3.76 ± 2.97 | 7.79 ± 5.22 | 11.36 ± 5.50 | 7.09 ± 5.47 | <0.001 |
| 10-year probability of Hip Fracture with BMD $\geq$3% | 61 (4.6%) | 275 (24.9%) | 579 (68.0%) | 915 (27.9%) | <0.001 |
| 10-year probability of major osteoporotic fracture with BMD $\geq$10% | 44 (3.3%) | 233 (21.1%) | 446 (52.4%) | 723 (22%) | <0.001 |
| 10-year probability of major osteoporotic fracture with BMD $\geq$20% | 7 (0.5%) | 33 (3%) | 69 (8.1%) | 109 (3.3%) | <0.001 |
| 10-year probability of hip Fracture with BMD $\geq$3% and/or 10-year probability of major osteoporotic fracture with BMD $\geq$10% | 68 (5.1%) | 311 (28.2%) | 588 (69.1%) | 967 (29.5%) | <0.001 |
| 10-year probability of hip Fracture with BMD $\geq$3% and/or 10-year probability of major osteoporotic fracture with BMD $\geq$20% | 61 (4.6%) | 275 (24.9%) | 579(68.0%) | 915 (27.9%) | <0.001 |
| Prior fractures (hip and vertebral) | 66 (5.0%) | 43 (3.9%) | 61 (7.2%) | 170 (5.2%) | 0.005 |
| Hip fracture | 2 (0.2%) | 11 (1.1%) | 32 (4.1%) | 45 (1.6%) | <0.001 |
| Vertebral fracture | 64 (4.8%) | 32 (2.9%) | 29 (3.4%) | 125 (3.8%) | 0.037 |
| Osteoporosis | | | | | |
| Femoral neck | 34 (2.6%) | 96 (8.7%) | 277 (32.5%) | 407 (12.4%) | <0.001 |
| Lumbar spine | 106 (8.0%) | 206 (18.7%) | 298 (35.0%) | 610 (18.6%) | <0.001 |
| Femoral neck and/or lumbar spine | 122 (9.2%) | 236 (21.4%) | 395 (46.4%) | 753 (23.0%) | <0.001 |

Abbreviations: BMD: Bone mineral density

**Table 3. History of fracture and 10-year probability of fractures with bone mineral density among participants with and without osteoporosis.**

|  | Osteoporosis at FN/LS (N = 753) | Non-Osteoporosis (N = 2527) | OR (95%CI) |
|---|---|---|---|
| Prior fractures (hip and vertebral) | 54 (7.2%) | 116 (4.6%) | 1.61 (1.15–2.24)** |
| Hip fracture | 28 (4.2%) | 17 (0.8%) | 5.55 (3.02–10.20)* |
| Vertebral fracture | 26 (3.5%) | 99 (3.9%) | 0.88 (0.56–1.36) |
| 10-yr probability of fractures with BMD |  |  |  |
| Hip fracture $\geq$ 3% | 583 (71.4%) | 377 (14.9%) | 14.27 (11.77–17.30)* |
| MOF $\geq$ 10% | 444 (59.0%) | 279 (11.0%) | 11.58 (9.56–14.02)* |
| MOF $\geq$ 20% | 89 (11.8%) | 20 (0.8%) | 16.80 (10.27–27.49)* |
| Hip $\geq$ 3% and/or MOF $\geq$ 10% | 552 (73.3%) | 415 (16.4%) | 13.98 (11.53–16.95)* |
| Hip $\geq$ 3% and/or MOF $\geq$ 20% | 538 (71.4%) | 377 (14.9%) | 14.27 (11.77–17.30)* |

*p <0.001

**p <0.05

Abbreviations: BMD: Bone mineral density; FN: Femoral neck; MOF: Major osteoporotic fracture; LS: Lumbar spine

CI 11.77–17.30 for MOF $\geq$ 20%) (**Table 3**). However, the likelihood of vertebral fracture was not significantly difference between participants with and without osteoporosis (OR 0.88, 95% CI 0.56–1.36, p = 0.66).

The proportion of participants with different risks of fractures stratified by age group is shown in **Fig 1**. A total of 170 (5.2%) participants among all participants had a history of hip and/or vertebral fracture. After excluding these participants with fracture history, 699 (21.3%) had osteoporosis, 355 (10.8%) had osteopenia with high risk of fracture (FRAX score of hip fracture $\geq$ 3% and/or MOF $\geq$ 20%), 1192 (36.3%) had osteopenia with low risk of fracture (FRAX score of hip fracture < 3% and MOF < 20%) and 864 (26.3%) had normal BMD. Taken together, a total of 1,224 (37.3%, 95%CI 35.7–39.0%) participants would be eligible for anti-osteoporosis therapy. The proportion of participants who would be eligible for anti-osteoporosis therapy increased significantly with age (15.5% versus 35.5% versus 73.6% for participants aged <60, 60–70 and >70 years, respectively, p <0.001). If the cut-off value of FRAX score for MOF $\geq$ 10% was used to determine high-risk individuals, 1,253 (38.2%) participants (15.7%, 37.5% and 74.1% for participants aged <60, 60–70 and >70 years, respectively) would be eligible for anti-osteoporosis therapy.

## Discussion

The current study is the largest study in Thailand aiming to determine the proportion of individuals who were eligible for osteoporotic therapy among 3,280 postmenopausal women aged 40–90 years old who were eligible for BMD measurement for osteoporosis screening. We found that 1224 (37.3%) of the 3280 participants (95%CI 35.7–39.0%) would be eligible for anti-osteoporosis therapy based on the NOF criteria. Among them, 170 were eligible based on the presence of history of fracture, 699 participants without fracture were eligible as they had osteoporosis and the rest 355 participants with osteopenia were eligible because they had 10-year probability of hip fracture >3% or MOF >20%. As expected, the proportion of participants who would be eligible for anti-osteoporosis therapy increased significantly with age from 15.5% among participants aged <60 years to 73.6% among participants aged >70 years.

This study may have clinical and public health implications as it provides age-stratified estimation of proportion of individuals who would be eligible for anti-osteoporotic therapy. The result will be a reference to help inform care gap in identification and treatment of individuals with high risk of fragility fracture in general population. In fact, the prevalence of individuals

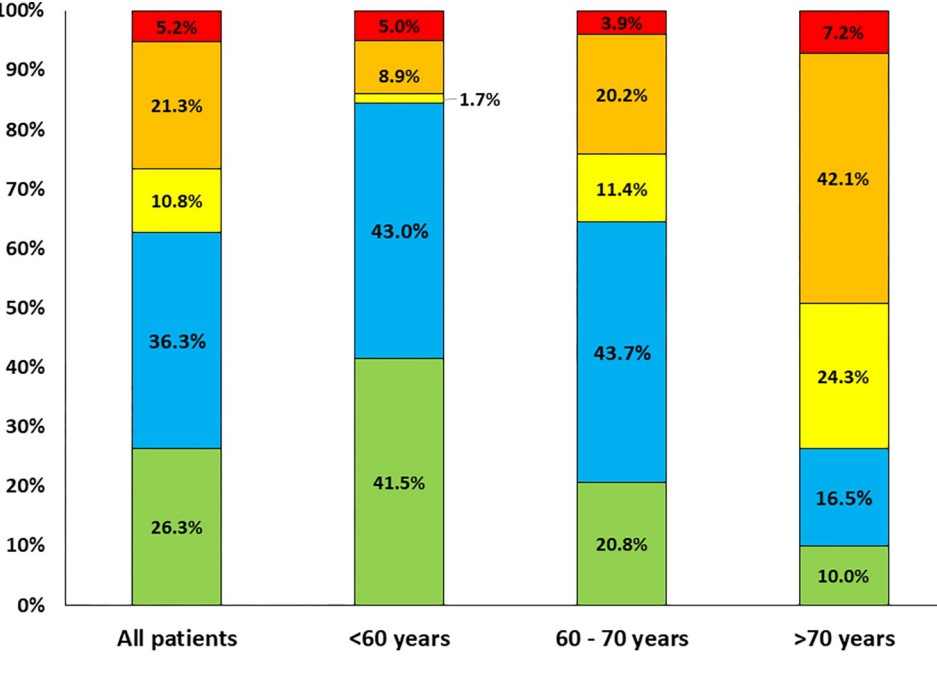

**Fig 1. Proportion of participants with different fracture risks by age group.** Osteoporosis and osteopenia were defined as T-score <-2.5 and T-score -1 ‒ -2.5 for LS or FN. BMD, respectively. Osteopenia with high risk of fracture was defined as participants with osteopenia and 10-year probability of hip fractures with BMD ≥ 3% and/or MOF with BMD ≥ 20%. Osteopenia with low risk of fracture was defined as participants with osteopenia and 10-year probability of hip fractures with BMD < 3% and MOF with BMD < 20%. Participants who would be eligible for anti-osteoporosis therapy included those with hip/vertebral fracture, osteoporosis and osteopenia with high risk of fracture. (37.3% for all participants; 15.5% for participants aged <60; 35.5% for participants aged 60–70 years; and 73. 6% for participants aged >70 years). Note that proportions of participants with osteoporosis, osteopenia with high risk of fracture, osteopenia with low risk of fracture and normal bone mineral density were calculated after excluding participants with hip/vertebral fracture. Abbreviations: BMD: Bone mineral density; FN: Femoral neck; LS: Lumbar spine; MOF: Major osteoporotic fracture.

who met treatment threshold based on FRAX score has been reported in the US population. However, such data in the Asian populations are relatively limited. Based on the US population-based Framingham study in 1,946 women, the proportion of women who met treatment criteria based on the 2008 NOF guideline was 41.1%. Concurrent with our observation, the proportion was much less among women aged <65 years (8.3%) compared with women aged >75 years (86.0%) [14]. An analysis of the National Health and Nutrition Examination Survey III of 1,754 women aged >50 years revealed that 37.4% met the 2008 NOF treatment threshold, and the proportion increased with age from 17.0% among those aged 50–59 years to 87.5% among those aged >80 years [15]. In Southeast Asia, a Vietnamese study in 1,421 women and 652 men aged 50 years or older, using the Thai version of FRAX, revealed that 49% of women and 35% of men would be eligible for osteoporotic therapy based on the NOF criteria [16].

While the NOF guideline has determined treatment threshold among patients with osteopenia as having a FRAX 10-year risk of at least 3% for hip fracture or at least 20% for MOF [3], the treatment threshold differs among different national guidelines given varying optimal cut-off values of FRAX based on local data [17]. In Thailand, a retrospective study conducted between 2008 and 2010 revealed that the original FRAX model with thresholds of ≥20% and ≥3% for MOF and hip fracture had moderate and low accuracy in predicting 10-year risk of

MOF (73% sensitivity, 63% specificity) and hip fracture (62% sensitivity, 60% specificity), respectively [18, 19]. A subsequent study in 2,872 postmenopausal Thai women used the receiver operating characteristic curve revealed the optimal FRAX thresholds for hip fracture with BMD was 4% (82.2% sensitivity, 78.6% specificity), and the optimal FRAX thresholds for MOF with BMD was 8.9% (87% sensitivity, 71% specificity) [20]. Given such data, we performed additional analysis in our cohort by lowering the FRAX cut-off value for MOF to 10%, which identified additional 29 patients (0.9%) who met treatment threshold.

It is of particular interest that there was no difference in the likelihood of vertebral fracture between participants with and without osteoporosis based on BMD criteria. One of the potential explanations is that presence of vertebral fracture can result in false elevation of measured LS BMD, which may have occurred in participants with vertebral fractures who were classified as not having osteoporosis.

The major strength of our study is the large sample size of 3,280 participants. This is also the first study to determine the prevalence of osteoporosis in postmenopausal women using both BMD T-score and Thai FRAX score criteria that represents the real-world practice. However, there are some limitations that should be acknowledged. First, data on vertebral fracture were obtained by self-report, which may have jeopardized the accuracy of ascertainment of history of fragility fracture. Secondly, most of participants in this study are from Northeastern region of Thailand, which may not represent the whole Thai population. Finally, we also lack the data on the proportion of treated patients and treatment outcome, which requires further studies.

In conclusion, the prevalence of osteoporosis in Thai postmenopausal women was 23.0%. A total of 37.3% would be eligible for anti-osteoporosis therapy based on the NOF criteria. The proportion increased significantly with age from 15.5% among participants aged <60 years to 73.6% among participants aged >70 years.

## Supporting information

**S1 Dataset.**
(XLSX)

## Author Contributions

**Conceptualization:** Piyachat Chanidkul, Dueanchonnee Sribenjalak, Chatlert Pongchaiyakul.

**Data curation:** Piyachat Chanidkul, Dueanchonnee Sribenjalak, Chatlert Pongchaiyakul.

**Formal analysis:** Piyachat Chanidkul, Dueanchonnee Sribenjalak, Nipith Charoenngam, Chatlert Pongchaiyakul.

**Investigation:** Piyachat Chanidkul, Dueanchonnee Sribenjalak, Nipith Charoenngam, Chatlert Pongchaiyakul.

**Methodology:** Piyachat Chanidkul, Dueanchonnee Sribenjalak, Nipith Charoenngam, Chatlert Pongchaiyakul.

**Visualization:** Nipith Charoenngam.

**Writing – original draft:** Piyachat Chanidkul, Dueanchonnee Sribenjalak, Chatlert Pongchaiyakul.

**Writing – review & editing:** Piyachat Chanidkul, Dueanchonnee Sribenjalak, Nipith Charoenngam, Chatlert Pongchaiyakul.

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
