## [Decision Letter · Decision Letter 0]

5 Sep 2022

PONE-D-22-14831The proportion of Thai postmenopausal women who would be eligible for anti-osteoporosis therapyPLOS ONE

Dear Dr. Charoenngam,

Thank you for submitting your manuscript to PLOS ONE. After careful consideration, we feel that it has merit but does not fully meet PLOS ONE’s publication criteria as it currently stands. Therefore, we invite you to submit a revised version of the manuscript that addresses the points raised during the review process.

Your manuscript has been assessed by one peer-reviewer and their report is appended below.  The reviewer comments that your manuscript needs further detail and/or clarification on several aspects, in particular the methodology and results section need further information, and the discussion section could be expand more upon the likelihood of vertebral fracture.   Please note that we have only been able to secure a single reviewer to assess your manuscript. We are issuing a decision on your manuscript at this point to prevent further delays in the evaluation of your manuscript. Please be aware that the editor who handles your revised manuscript might find it necessary to invite additional reviewers to assess this work once the revised manuscript is submitted. However, we will aim to proceed on the basis of this single review if possible. 

We look forward to receiving your revised manuscript.

Kind regards,

Maria Elisabeth Johanna Zalm, Ph.D

Editorial Office

PLOS ONE

Journal Requirements:

Reviewers' comments:

Reviewer's Responses to Questions

**Comments to the Author**

1. Is the manuscript technically sound, and do the data support the conclusions?

Reviewer #1: Yes

2. Has the statistical analysis been performed appropriately and rigorously? 

Reviewer #1: Yes

3. Have the authors made all data underlying the findings in their manuscript fully available?

Reviewer #1: Yes

4. Is the manuscript presented in an intelligible fashion and written in standard English?

Reviewer #1: Yes

5. Review Comments to the Author

Reviewer #1: The strength of study is large sample size. The manuscript is overall well written and informative. The study was properly designed and executed. The finding can help inform treatment guidelines in Thailand. However, there are some issues that I would like to invite the authors to comment on.

1. Please clarify the define of postmenopausal in this study? How many women aged 40-50 were there? Did all of them have natural or premature/induced menopause?

2. In the methods, please describe more details about prior fractures (trauma or non-trauma)?

3. The point estimate of the proportion of Thai menopausal women who would be eligible for treatment was given (37.3%). Please provide the related confidence intervals, quantifying the potential uncertainty of the finding which would make the interpretation stronger and more convincingly.

4. The authors defined “high-risk osteopenia” (as participants with osteopenia and 10-year probability of hip fractures with BMD ≥ 3% and/or MOF with BMD ≥ 20%.), but I think the term “high risk of fracture” is more appropriate.

5. Please discuss more about the likelihood of vertebral fracture was not significantly difference between participants with and without osteoporosis

6. PLOS authors have the option to publish the peer review history of their article (what does this mean?). If published, this will include your full peer review and any attached files.

Reviewer #1: **Yes: **Lan T. Ho-Pham

---

## [Decision Letter · Decision Letter 1]

15 Dec 2022

The proportion of Thai postmenopausal women who would be eligible for anti-osteoporosis therapy

PONE-D-22-14831R1

Dear Dr. Charoenngam,

We’re pleased to inform you that your manuscript has been judged scientifically suitable for publication and will be formally accepted for publication once it meets all outstanding technical requirements.

Kind regards,

Inge Roggen, M.D., Ph.D.

Academic Editor

PLOS ONE

Additional Editor Comments (optional):

Reviewers' comments:

Reviewer's Responses to Questions

**Comments to the Author**

1. If the authors have adequately addressed your comments raised in a previous round of review and you feel that this manuscript is now acceptable for publication, you may indicate that here to bypass the “Comments to the Author” section, enter your conflict of interest statement in the “Confidential to Editor” section, and submit your "Accept" recommendation.

Reviewer #1: All comments have been addressed

Reviewer #2: All comments have been addressed

2. Is the manuscript technically sound, and do the data support the conclusions?

Reviewer #1: Yes

Reviewer #2: Yes

3. Has the statistical analysis been performed appropriately and rigorously? 

Reviewer #1: Yes

Reviewer #2: Yes

4. Have the authors made all data underlying the findings in their manuscript fully available?

Reviewer #1: Yes

Reviewer #2: Yes

5. Is the manuscript presented in an intelligible fashion and written in standard English?

Reviewer #1: Yes

Reviewer #2: No

6. Review Comments to the Author

Reviewer #1: The authors have addressed all the issues raised. I do not have any more questions. The manuscript now is suitable to PLOS ONE.

Reviewer #2: Piyachat Chanidkul et al have done a study looking at the proportion of postmenopausal Thai women eligible for treatment for osteoporosis.

Strengths: it is a large study of 3280 women and the study is well described.

Weaknesses:

1 Using Pubmed and Scholar I found many papers about ostoporosis in Asian women and as such this paper does not contain new information.

2 The authors have just used the NOF criteria to identify women eligible for therapy. There are many studies already published with similar outcomes.

3 The authors should have used multi-regression analysis to analyze the relationship between variables.

7. PLOS authors have the option to publish the peer review history of their article (what does this mean?). If published, this will include your full peer review and any attached files.

Reviewer #1: **Yes: **Lan T. Ho-Pham

Reviewer #2: No

---

## [Editor Report · Acceptance letter]

25 Jan 2023

PONE-D-22-14831R1 

The proportion of Thai postmenopausal women who would be eligible for anti-osteoporosis therapy 

Dear Dr. Charoenngam:

I'm pleased to inform you that your manuscript has been deemed suitable for publication in PLOS ONE. Congratulations! Your manuscript is now with our production department. 

Kind regards, 

on behalf of

Dr. Inge Roggen 

Academic Editor

PLOS ONE